# Formation of metastable phases by spinodal decomposition

Ricard Alert[1,2], Pietro Tierno[1,2,3] & Jaume Casademunt[1,2]

Metastable phases may be spontaneously formed from other metastable phases through nucleation. Here we demonstrate the spontaneous formation of a metastable phase from an unstable equilibrium by spinodal decomposition, which leads to a transient coexistence of stable and metastable phases. This phenomenon is generic within the recently introduced scenario of the landscape-inversion phase transitions, which we experimentally realize as a structural transition in a colloidal crystal. This transition exhibits a rich repertoire of new phase-ordering phenomena, including the coexistence of two equilibrium phases connected by two physically different interfaces. In addition, this scenario enables the control of sizes and lifetimes of metastable domains. Our findings open a new setting that broadens the fundamental understanding of phase-ordering kinetics, and yield new prospects of applications in materials science.

[1] Departament de Física de la Matèria Condensada, Universitat de Barcelona, Avinguda Diagonal 647, 08028 Barcelona, Spain. [2] Universitat de Barcelona Institute of Complex Systems (UBICS), Universitat de Barcelona, 08028 Barcelona, Spain. [3] Institut de Nanociència i Nanotecnologia, Universitat de Barcelona, 08028 Barcelona, Spain. Correspondence and requests for materials should be addressed to R.A. (email: ricardaz@ecm.ub.edu) or to J.C. (email: jaume.casademunt@ub.edu).

Upon a quench, namely a sudden change of the external conditions, a system is initially in a nonequilibrium state. The processes of relaxation to the new equilibrium state may be complex, specially if a phase transition boundary is crossed and more than one phase is locally stable. The corresponding phase-ordering processes and their kinetics have been intensively studied for decades and are now a classical topic of nonequilibrium physics. Indeed, phase-ordering kinetics is central to understand and control domain formation in a wide range of materials, ranging from liquid mixtures or metal alloys to structural or magnetic domains in solids, through liquid crystals, polymers and many soft-matter systems[1].

The dynamics of phase transitions often involves metastable phases, namely states that are only transiently stable, since they relax to the actual equilibrium by a finite-size perturbation. Examples are ubiquitous and include diamond or supercooled liquid water, which are metastable, respectively, to graphite and ice at room pressure. A given phase may become metastable upon a change of thermodynamic variables, such as temperature, pressure or magnetic field, that modifies its relative stability, such as when liquid water is supercooled. Subsequently, a transition to the stable equilibrium phase occurs typically via nucleation, which requires overcoming an energy barrier to form a growing nucleus of the final phase. In contrast, if the quench is such that the initial phase is in unstable equilibrium, new equilibrium phases are spontaneously generated by the relaxation dynamics. By this process, known as spinodal decomposition, infinitesimal fluctuations directly grow to give rise to domains of the final coexisting phases[2–5].

Metastable phases may be generated *de novo* through nucleation from other preexisting metastable phases, such as supercooled water giving rise to metastable structures of ice. According to Ostwald's step rule[6], this will occur when the nucleation kinetics of the stable phase is slower than that of an intermediate metastable phase. On the other hand, nonequilibrium metastable states such as gels may form via a dynamic arrest of a spinodal decomposition process[7,8]. However, to our knowledge, metastable equilibrium phases, namely metastable states of possible equilibrium phases of the system, have never been observed to appear spontaneously by spinodal decomposition, this is from an unstable equilibrium phase. Here we predict and experimentally verify the direct, spontaneous formation of a metastable equilibrium phase upon a quench into an unstable equilibrium.

This phenomenon is observed in a solid–solid transition of a two-dimensional (2D) colloidal crystal made of paramagnetic particles. Colloidal systems have proven to be very useful experimental models for studying the kinetics of phase transitions[9]. Specifically, several aspects of the dynamics of solid–solid transitions were revealed by studies on colloidal crystals[10], including the appearance of long-lived metastable structures[11–13], usually through displacive transformations. Indeed, diffusive nucleation was only recently observed in colloidal crystals, in the form of a two-stage, liquid-mediated process[14,15] that was later found in a metal[16].

From a fundamental point of view, the universal features of phase-ordering processes are usually captured by time-dependent continuum models based on coarse-grained free energy functionals[2–5,17]. Recently, these classical field models are being revisited to formulate a theory for phase separation in active systems[18,19]. However, the fundamental theory of phase separation in traditional, non-active systems has remained essentially unchanged for decades. Herein, we report a battery of new phase-ordering phenomena that have no counterpart within the classical theory. Our results stem from a recently introduced, nonstandard scenario of phase transitions, the so-called landscape-inversion phase transitions (LIPT)[20], where the periodic energy landscape of the system can be inverted by changing a single parameter.

Among the new phenomena, we highlight the existence of an asymmetric spinodal decomposition whereby the system phase separates into two coexisting equilibrium phases of different relative stability. This process leads to the aforementioned formation of the metastable domains. We also predict that these domains are subsequently eliminated by a front propagation mechanism, thus differing from the self-similar domain coarsening that usually follows spinodal decomposition. Moreover, we also show that the range of sizes and lifetimes of the metastable domains, and thus the overall phase transition kinetics, can be externally controlled by a magnetic field. Finally, we further reveal the possibility that two coexisting stable phases are simultaneously connected by two distinct interfaces, with different physical properties such as interfacial tension.

## Results

**The landscape-inversion phase transition.** A phase transition scenario based on a complete inversion of the energy landscape was recently discovered in a 2D crystal of paramagnetic colloidal particles on top of a periodic magnetic substrate[20], and later found in a suspension of magnetic and nonmagnetic particles within a ferrofluid of tunable susceptibility[21]. In the former realization, particles arrange along parallel lines following the domain walls of a striped substrate. At these lines, the substrate generates a magnetic field $\pm H_s$ that magnetizes particles on consecutive lines in opposite directions (Fig. 1a). Then, when a uniform external magnetic field $H$ is applied, particles on consecutive lines acquire magnetic dipoles $m_i \propto H + H_s$ and $m_j \propto H - H_s$. Thus, the dipolar interaction between these particles yields a contribution $U_{ij} \propto m_i m_j \propto H_s^2 - H^2$ to the energy. Consequently, the application of an external field $H > H_s$ causes the energy landscape to globally invert. This induces a structural phase transition in the crystal, with the lattice angle $\alpha$ as (nonconserved) order parameter, which is accompanied by a magnetic transition from an antiferromagnetic to a ferromagnetic order (Fig. 1). Moreover, without crossing the phase transition boundary at $H = H_s$, the external field tunes the relative stability of the different structures, and hence their dynamics, without modifying their crystalline order $\alpha$, which can be independently tuned by changing the density of particles[20].

We remark that, for $H < H_s$, the equilibrium angle $\alpha_a$ is equivalent to $-\alpha_a$ (Fig. 1c). This is because, in the corresponding structure, particles on one line are equidistant from the four neighbouring particles on the two nearest lines (sketch in Fig. 1a). Therefore, the structures with lattice angles $\alpha_a$ and $-\alpha_a$ are exactly the same, and hence the identification $\alpha_a \leftrightarrow -\alpha_a$. In contrast, for $H > H_s$, the two equilibrium structures, with opposed lattice angles $\alpha_b$ and $-\alpha_b$ (Fig. 1c), correspond to different particle arrangements (sketch in Fig. 1b).

**Formation of metastable domains by spinodal decomposition.** We first focus on the phase-ordering kinetics associated to the LIPT by suddenly switching off an external magnetic field $H = 3H_s/2$. This quench forces the crystal to transit from the rhomboidal structure with $\alpha_b \approx 7°$ (Fig. 1b) to that with $\alpha_a \approx 25°$ (Fig. 1a). We monitor the dynamics of the structural rearrangement via the time evolution of the radial distribution function $g(r)$ of the crystal, as shown in Fig. 2. Different crystalline lattices are distinguished by their corresponding peaks in the $g(r)$. In Fig. 2a, the initial $\alpha_b$ crystal (Fig. 1b) is distinguished by the presence of secondary peaks at $r \sim 1.4\lambda/2$ and $r \sim 2.25\lambda/2$ (blue). In turn, the final $\alpha_a$ crystal (Fig. 1a) features a

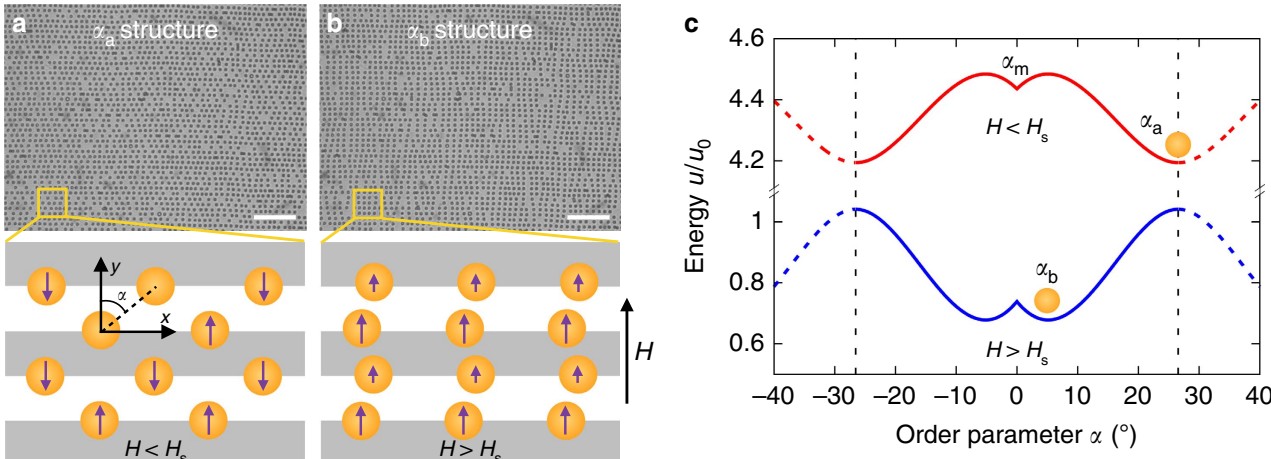

**Figure 1 | Landscape-inversion phase transition in dipolar colloids on a periodic magnetic substrate.** (**a,b**) Experimental image and sketch of the dipolar colloidal crystal at external magnetic fields $H$ lower and higher than the substrate field $H_s = 13\,\mathrm{kA\,m^{-1}}$. Particles assemble into lines, on top of the walls between oppositely magnetized domains (white and grey) of the substrate, with spatial period $\lambda = 2.6\,\mu m$. The structural order is described by the lattice angle $\alpha$. Scale bars, 20 μm. (**c**) The energy landscape of the system completely inverts for $H > H_s$. This induces a transition from the $\alpha_a$ to the $\alpha_b$ structure in the crystal[20]. A metastable equilibrium structure, $\alpha_m$, exists for $H < H_s$. Dashed lines illustrate the identification $\alpha_a \leftrightarrow -\alpha_a$, since these angles correspond to the same structure (see text). The energy scale is $u_0 \equiv \mu_0 \chi^2 a^3 H_s^2$, with $\chi \sim 1$ the magnetic susceptibility of the particles of radius $a = 0.5\,\mu m$.

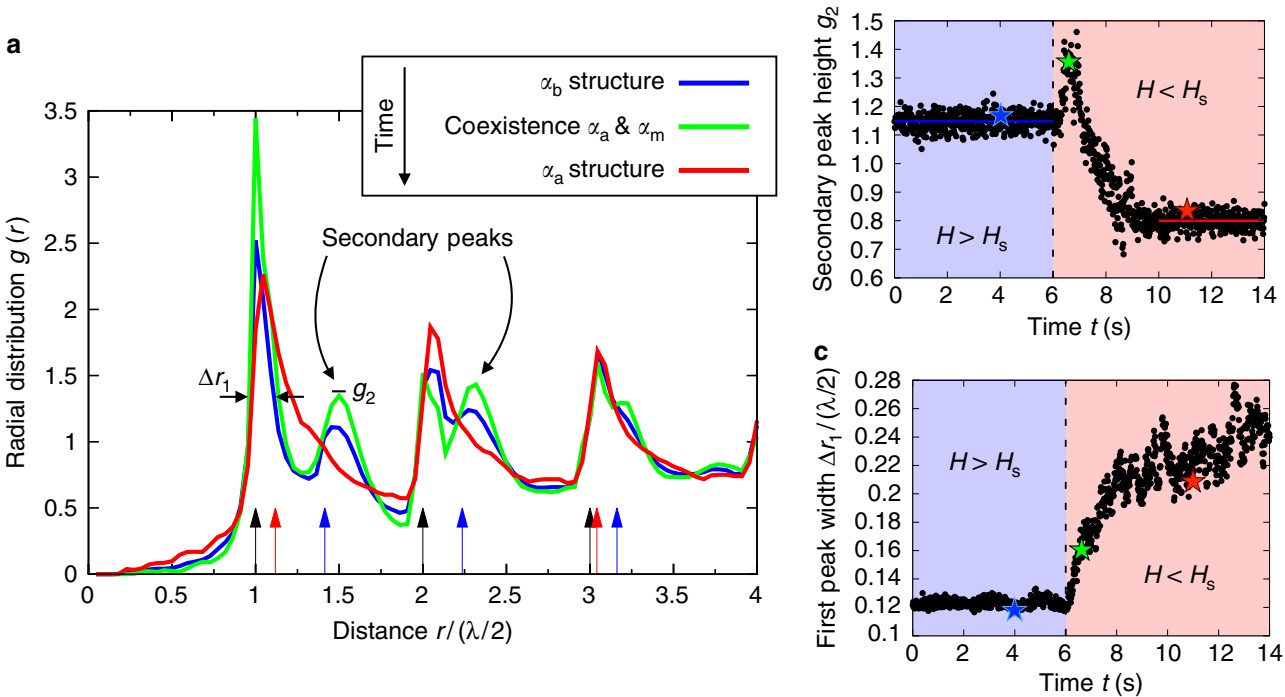

**Figure 2 | Formation of a metastable phase.** (**a**) Radial distribution function $g(r)$ of the system before ($t = 4\,s$, blue), during ($t = 6.57\,s$, green), and after ($t = 11\,s$, red) the phase-ordering process induced by a quench of the magnetic field from $H = 3H_s/2$ to $H = 0$ at $t = 6\,s$. The theoretical positions of the main peaks of the initial and final equilibrium structures are indicated by blue and red arrows, respectively, with black arrows standing for peaks common to both lattices. The distance is divided by the spatial semiperiod of the substrate, $\lambda/2 = 1.3\,\mu m$. (**b**) Time evolution of the height of the secondary peak of the $g(r)$. Its transient increase right after the quench is due to the spontaneous formation of a metastable rectangular structure with $\alpha_m = 0$, which disappears afterwards. Colour lines indicate the average height of the peak at equilibrium, both before and after the quench. Colour stars correspond to the three radial distribution functions in **a**. (**c**) Time evolution of the width of the first peak of the $g(r)$ at a height $g = 1.3$. After the quench, this peak widens due to the appearance of a second peak of the final lattice very close to the primary peak, which is not resolved experimentally. The widening, this is the formation of the final equilibrium phase, occurs on the same characteristic time as the disappearance of the metastable phase shown in **b**. Therefore, the metastable domains are eliminated in favour of the stable crystalline structure. All quantities are averaged over 15 realizations.

wider first peak and no secondary peaks (red). The theoretical positions of the main peaks of both structures are indicated by arrows in Fig. 2a. From them, we infer that the widening of the first peak in the $\alpha_a$ structure indeed results from the appearance of a second peak at $r \sim 1.1\lambda/2$, which cannot be resolved from the primary one within the experimental resolution.

However, Fig. 2a also reveals that the structural transition does not occur via a homogeneous relaxation but that it rather involves a nontrivial phase-ordering process. This follows from the fact that the secondary peaks of the initial structure first increase in height (green) before disappearing (see also Fig. 2b). This corresponds to the transient formation of the metastable rectangular structure with $\alpha_m = 0$ (Fig. 1c), which contributes peaks at $r = \sqrt{2}\lambda/2$ and $r = \sqrt{5}\lambda/2$. However, the stable $\alpha_a$ structure also starts forming right after the quench, as indicated by the widening of the first peak of the $g(r)$ (Fig. 2c). Therefore, both phases transiently coexist for $\sim 2$ s, after which the system reaches the final homogeneous $\alpha_a$ phase.

The theoretical model of the LIPT (Fig. 1c) allows to predict the phase-ordering processes associated to the inversion of the energy landscape. A simple quench from $H > H_s$ to $H < H_s$ leaves the system at an unstable equilibrium state, from which it phase separates into domains of two locally stable phases. However, in contrast to standard spinodal decomposition[2–5], the two coexisting phases have different relative stability. Hence, we term this phase separation process 'asymmetric spinodal decomposition'. Thereby, half of the system initially evolves towards a metastable phase that was not present in the initial condition. Therefore, in the light of the model, the experimental data conclusively demonstrate the direct

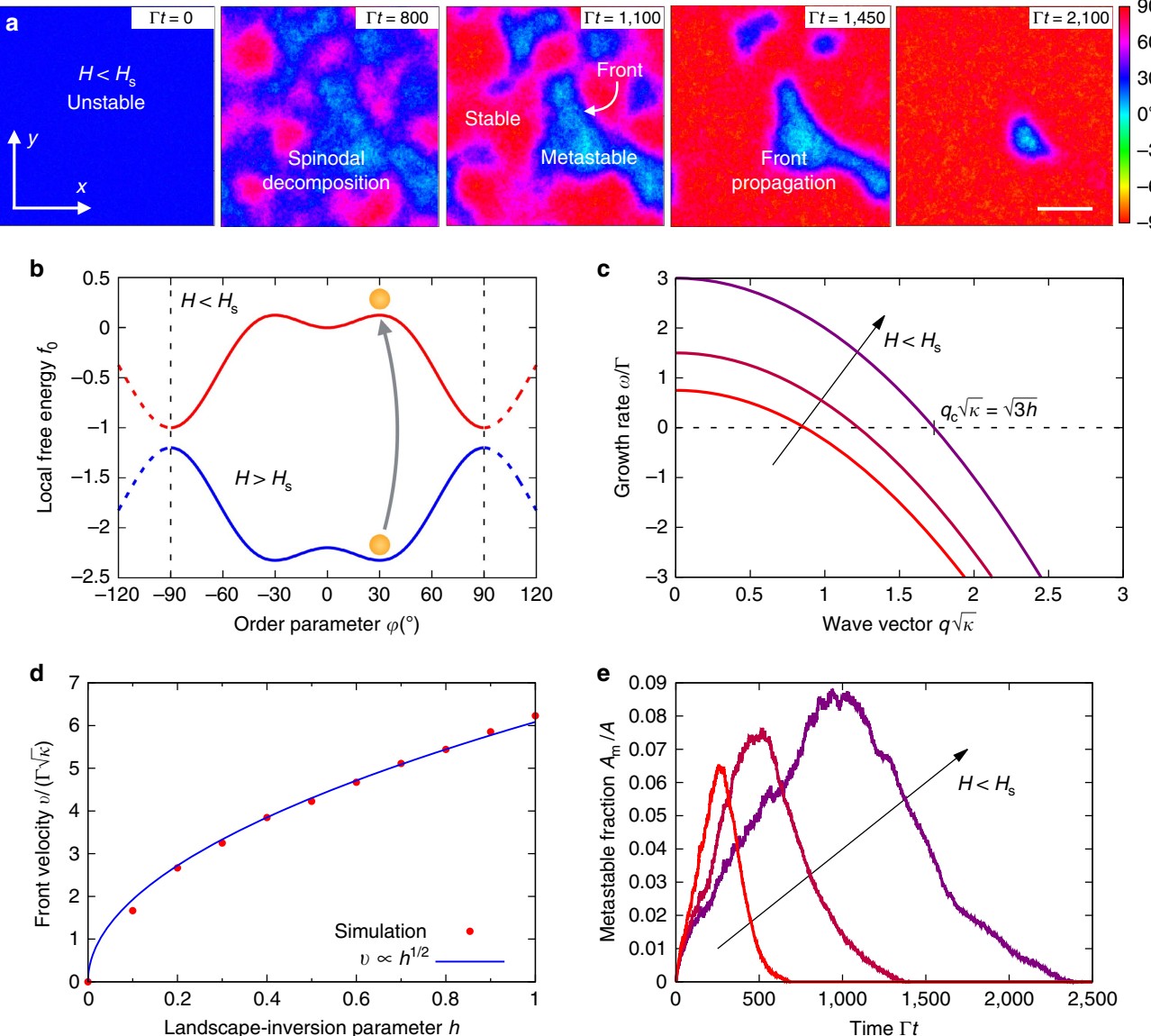

**Figure 3 | Formation of a metastable phase by spinodal decomposition and front propagation.** (**a**) Simulation snapshots of the phase-ordering process resulting from a quench from $H > H_s$ to $H < H_s$ (see Supplementary Movie 1). Colour indicates the order parameter field $\varphi(\mathbf{r}, t)$ of the model. The system, initially at the unstable state $\varphi = 30°$ (dark blue), undergoes an asymmetric spinodal decomposition that spontaneously generates both stable $\varphi = 90°$ (red) and metastable $\varphi = 0°$ (light blue) domains. The latter are subsequently invaded by the former via propagating fronts. $\Gamma^{-1}$ and $\sqrt{\kappa}$ define the time and length units in the simulations, respectively. Scale bar, $50\sqrt{\kappa}$. (**b**) Model energy landscape of the LIPT, equation 2. The dashed part illustrates the periodicity of the potential. (**c**) Dispersion relation of the unstable state, equation 4, for different values of the magnetic field. The field controls the region of unstable modes $q < q_c$ and thus the range of sizes of the forming domains. (**d**) Velocity of planar fronts as a function of the landscape-inversion parameter $h \equiv 1 - H^2/H_s^2$ from simulations, with a fit of the predicted scaling $v \propto h^{1/2}$. (**e**) Time evolution of the area fraction covered by the metastable state for different values of the magnetic field. This graph shows the formation and subsequent elimination of the metastable state, with the overall phase transition dynamics controlled by the external magnetic field.

spontaneous formation of a metastable phase by spinodal decomposition.

Hitherto, metastable phases could only form *de novo* from other metastable phases, either through diffusive nucleation or displacive transformations. Indeed, a mechanism for the formation of metastable domains upon the nucleation of the stable phase was proposed based on a dynamical instability of the fronts propagating from stable into unstable states, both for nonconserved[22–24] and conserved[25,26] order parameters. In any case, the formation of the metastable phases was not generic but depended on the depth or rate of the quench. In contrast, within the LIPT scenario, metastable domains are spontaneously formed by spinodal decomposition, thus directly from the unstable state, simultaneously to the formation of stable domains. In other words, metastable domains are naturally formed at the initial stages of the phase separation process, as a direct consequence of the energy landscape inversion. In addition, their appearance is completely generic, independent of the quench depth.

To further investigate the novel phase-ordering processes of the LIPT, we formulate a time-dependent Ginzburg–Landau-like model for its dynamics (see Methods). Stochastic simulations of such a model clearly illustrate the spontaneous formation of metastable domains upon the inversion of the energy landscape (Fig. 3a,b). The dispersion relation of the unstable state, plotted in Fig. 3c, gives the minimal size of the forming domains, $\sim 2\pi/q_{\mathrm{c}}$, which is controlled by the magnetic field at which the quench is performed.

**Front propagation**. The generated metastable domains coexist with globally stable ones, and, therefore, they are subsequently invaded and eliminated by fronts of the stable state. Consequently, the late stages of the phase-ordering process do not proceed by a self-similar coarsening as in usual spinodal decomposition[3–5]. Thus, domain dynamics is not governed by interfacial curvature but rather by front propagation, and hence by the free energy difference between the stable and metastable phases[27–29]. This enables the external control of the domain dynamics by means of the magnetic field.

The dependence of the speed of the fronts on the value of the magnetic field upon the quench can be deduced from the dynamics of the order parameter field[28,29]. Neglecting curvature corrections, the interface speed, width and tension are thereby predicted to scale as $v \propto h^{1/2}$, $\delta \propto h^{-1/2}$, and $\sigma \propto h^{1/2}$, respectively (see Methods). Here we have defined the landscape-inversion parameter $h \equiv 1 - H^2/H_{\mathrm{s}}^2$, which changes sign at the transition

point. Then, we measure the speed of planar fronts in simulations for several values of $h$. The numerical results agree with the predicted scaling, as shown in Fig. 3d.

We note that the scaling $v \propto h^{1/2}$ contrasts with the prediction of a linear scaling $v \propto \epsilon$ of the front speed with the distance from the transition point, $\epsilon \equiv (T - T_{\mathrm{c}})/T_{\mathrm{c}}$, for first-order phase transitions[30]. For the LIPT, $h$ directly measures the distance from the transition point at $h = 0$. In fact, a square-root scaling $v \propto |\epsilon|^{1/2}$ like the one we find was predicted for fronts associated to second-order transitions[30]. Thus, despite being discontinuous, the LIPT shares some dynamical properties with second-order phase transitions.

Finally, since the external magnetic field controls both the range of sizes of the generated domains and their rate of disappearance, it actually tunes the overall phase transition dynamics. Higher fields, closer to the transition threshold $H_{\mathrm{s}}$, imply a wider distribution of domain sizes (Fig. 3c) and slower propagating fronts, so that more metastable phase is generated and it lives longer. This is shown in Fig. 3e, which reports the fraction of area covered by the metastable state in simulations at different magnetic fields.

Last, it is also worth mentioning that, in principle, the inversion of the energy landscape could be useful as a generic probe for front propagation problems (see Supplementary Note).

**Phase coexistence with two physically different interfaces**. We next consider the phase-ordering process upon the opposite quench, from $H < H_{\mathrm{s}}$ to $H > H_{\mathrm{s}}$. Again, as illustrated in Fig. 4a, the inversion of the energy landscape from the equilibrium state $\varphi = 90°$ at $H < H_{\mathrm{s}}$ leaves the system at an unstable state, from which it undergoes spinodal decomposition. However, this process is symmetric for the present situation, leading to two equilibrium phases, $\varphi = \pm 30°$, with equal energy. As a consequence, interfaces between these two phases can not propagate as fronts, and they only move under curvature. Their speed is locally proportional to their curvature, following the Allen–Cahn law[31]. This behaviour leads to a usual curvature-driven coarsening process characterized by a scaling regime of the typical domain size[2–5,31,32] $R(t) \propto t^{1/2}$.

Now, because of the periodicity of the free energy, the two equilibrium phases are indeed connected through two distinct interfaces, with different profiles and interfacial tensions (Fig. 4a). Remarkably, the system forms only the most energetic of both interfaces when quenched from the equilibrium state $\varphi = 90°$ at $H < H_{\mathrm{s}}$. In contrast, if the system presents stable-metastable

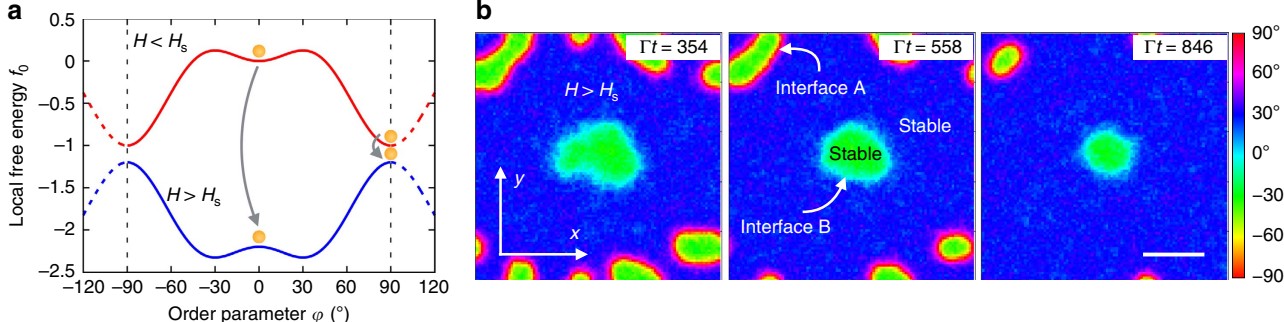

**Figure 4 | Two physically different interfaces connect the two equilibrium phases.** (**a**) Model energy landscape of the LIPT, equation 2. The dashed part illustrates the periodicity of the potential. (**b**) Simulation snapshots of the phase-ordering process that results from a quench from $H < H_{\mathrm{s}}$ to $H > H_{\mathrm{s}}$ (see Supplementary Movie 2). Colour indicates the order parameter field $\varphi(\mathbf{r}, t)$ of the model. From an initial coexistence of a stable $\varphi = 90°$ (red) and a metastable $\varphi = 0°$ (light blue) domains, the system undergoes a double spinodal decomposition. This leads to the formation of two physically different interfaces (A and B) connecting the two equilibrium phases $\varphi = \pm 30°$ (dark blue and green, respectively). $\Gamma^{-1}$ and $\sqrt{\kappa}$ define the time and length units in the simulations, respectively. Scale bar, $50\sqrt{\kappa}$.

## Landscape-inversion scenario

### Order parameter

|  | Nonconserved | Conserved |
|---|---|---|
| **Quench To h > 0** | Asymmetric spinodal decomposition + Front propagation | Asymmetric spinodal decomposition + Diffusive coarsening ($t^{1/3}$ scaling) |
|  | · Transient stable/metastable coexistence | · Permanent stable/metastable coexistence |
| **Quench To h < 0** | (Double) Symmetric spinodal decomposition + Curvature-driven coarsening ($t^{1/2}$ scaling) | Double symmetric spinodal decomposition + Diffusive coarsening ($t^{1/3}$ scaling) |
|  | · Two equilibrium phases connected by two physically different interfaces | · Two equilibrium phases connected by two physically different interfaces |

## Classical models

### Order parameter

|  | Nonconserved (model A) | Conserved (model B) |
|---|---|---|
| **Critical** | Spinodal decomposition + Curvature-driven coarsening ($t^{1/2}$ scaling) | Spinodal decomposition + Diffusive coarsening ($t^{1/3}$ scaling) |
|  | · Bicontinuous patterns | · Bicontinuous patterns |
| **Off-critical — Below spinodal** | Homogeneous relaxation to the broken-symmetry phase No phase separation | Spinodal decomposition + Diffusive coarsening ($t^{1/3}$ scaling) |
|  |  | · Droplet patterns |
| **Off-critical — Below binodal** | Nucleation + Front propagation | Nucleation + Diffusive coarsening ($t^{1/3}$ scaling) |

**Figure 5 | Comparison of the phase-ordering kinetics of the LIPT and classical scenarios.** Initial stages of phase separation (domain formation) are indicated in red, and late stages (domain growth) are indicated in blue. Key properties of some processes are specified (shaded). Details are given in the Supplementary Discussion. Only the results of quenches to temperatures below critical, $T < T_c$, are shown for the classical models, since quenches to $T > T_c$ lead to a homogeneous relaxation without phase separation. In the LIPT scenario, phase separation processes occur at both sides of the transition.

coexistence when quenched to $H > H_s$, it naturally undergoes a double spinodal decomposition leading to the formation of both interfaces, as illustrated in Fig. 4b. Simulations also show that the most energetic interface disappears in favour of the least energetic one when both get into contact (Supplementary Movie 3).

## Discussion

Our findings open a new scenario of phase-ordering kinetics, associated to the LIPT, in which several unexpected phenomena take place. In particular, these include a new mechanism to form metastable phases, namely by spinodal decomposition. By this mechanism, metastable phase formation is robust within the LIPT scenario, occurring regardless of the amplitude or the rate of the quench. Remarkably, the LIPT never leads to nucleation despite the existence of a metastable state in the free energy. A summary of the phase-ordering processes allowed by this nonstandard scenario is provided in Fig. 5a. There, the case of a conserved order parameter has also been included for completeness (see Supplementary Discussion for details). Additionally, a concise summary of the phase-ordering processes associated to the classical A and B models of phase transition dynamics[5] is given in Fig. 5b for comparison (see Supplementary Discussion).

While we experimentally demonstrate the unstable-to-metastable phase change, the rest of our predictions remain to be experimentally verified due to limitations of our setup, such as the relatively small system size, the presence of vacancies in the crystals and the lack of precise control of the in-line particle density. However, the LIPT scenario has been already realized in another physical system[21]. Thus, searching for other systems that broaden the experimental possibilities or finding a LIPT with conserved order parameter remain appealing open challenges. In this sense, our results may open new research avenues in the field of phase transition dynamics and may foster, for instance, the exploration of nonstandard routes to phase separation.

With regard to colloidal materials, in addition to enabling novel routes for solid–solid transitions, our system also provides external magnetic control over the phase-ordering dynamics of 2D crystals[10]. Understanding and controlling the ordering and relative stability of different crystalline structures could indeed be relevant for applications of colloidal crystals themselves[33,34], such as photonic band-gap materials, but also of atomic alloys in the nanotechnological domain[35,36]. In this respect, our work could provide a basis for the search of new self-assembly strategies, in

particular related to the possibility of tuning the spatial organization of crystalline structures in 2D materials.

## Methods

**Experiments.** The periodic magnetic substrate is a uniaxial ferrite garnet film grown by liquid phase epitaxy. An aqueous suspension of paramagnetic colloidal particles (Dynabeads Myone) is deposited on top. The external magnetic field is generated by custom-made coils. Particle positions are tracked by a custom-made software from video microscopy recordings at 60 Hz over an area of $140 \times 105 \,\mu m^2$.

**Dynamical field model of the LIPT.** We build a Ginzburg–Landau-like model for the dynamics of the LIPT by formulating a coarse-grained free energy functional of a scalar order parameter $\varphi(\mathbf{r}, t)$[37]:

$$F[\varphi] = u \int_{\Omega} \left( f_0[\varphi] + \frac{\kappa}{2} (\nabla \varphi)^2 \right) \mathrm{d}^d \mathbf{r}. \tag{1}$$

where $u$ and $\kappa$ are phenomenological parameters governing the energy scale and the spatial coupling, respectively.

The dimensionless local free energy density $f_0$ must capture the essential features of the actual potential of the LIPT (Fig. 1c). Therefore, $f_0$ must feature, at least, a stable and a metastable state, and it must allow for a complete inversion under change of a control parameter. The order parameter must have the topology of an angle, hence identifying the two extremum values of its range ($\alpha_a \leftrightarrow -\alpha_a$ in Fig. 1c, and $90° \leftrightarrow -90°$ in this model, Fig. 3b). A model free energy including all these ingredients is

$$f_0[\varphi] = h \sin^2 \varphi \left( 1 - A \sin^2 \varphi \right), \tag{2}$$

where $h$ and $A$ are two control parameters. Here we have defined the landscape-inversion parameter $h \equiv 1 - H^2/H_s^2$, while the role of the particle density is played by $A$, which we take to be $A = 2$. Indeed, $0 < h \leq 1$ corresponds to $0 \leq H < H_s$, and $h < 0$ corresponds to $H > H_s$, so that $h$ changes sign to invert the energy landscape. In turn, $A > 1$ controls the relative stability of the stable $\varphi = 90°$ and metastable $\varphi = 0$ states for $h > 0$, and the energy barrier between the two degenerated states at $\varphi = \pm \arcsin \sqrt{1/(2A)} = 30°$ for $h < 0$.

Then, the dynamics of the nonconserved order parameter is specified by the time-dependent Ginzburg–Landau equation

$$\frac{\partial \varphi}{\partial t} = -\frac{\Gamma}{u} \frac{\delta F[\varphi]}{\delta \varphi} + \xi = \Gamma \left( -\frac{\partial f_0}{\partial \varphi} + \kappa \nabla^2 \varphi \right) + \xi, \tag{3}$$

where $\Gamma$ is a kinetic coefficient and $\xi(\mathbf{r}, t)$ is a Gaussian white noise field with $\langle \xi(\mathbf{r}, t) \xi(\mathbf{r}', t') \rangle = 2D\delta(\mathbf{r} - \mathbf{r}')\delta(t - t')$. A linear stability analysis of the unstable states $\varphi = \pm 30°$ at $H < H_s$ leads to their dispersion relation:

$$\omega(q) = \Gamma \left( 3h - \kappa q^2 \right), \tag{4}$$

where $q$ is the modulus of the wave vector.

Finally, an extended model including the possibility of liquid interfaces between crystalline regions is introduced in the Supplementary Method (Supplementary Fig. 1), which is closer to the experimental situations of Fig. 1 or liquid-mediated transitions[14].

**Front dynamics**. The properties and motion of the interfaces follow from the order parameter dynamics[28,29]. Particularly, a closed equation for the profile of a steadily moving flat interface can be derived from equation 3. In the comoving frame of reference of a planar front advancing at a constant velocity $\mathbf{v}$, the evolution of the order parameter can be written as $\partial_t \varphi = -\mathbf{v} \cdot \nabla \varphi$, so that the time-dependent Ginzburg–Landau equation (equation 3) becomes

$$-v\frac{\mathrm{d}\varphi}{\mathrm{d}z} = \Gamma\left(-\frac{\partial f_0}{\partial \varphi} + \kappa\frac{\mathrm{d}^2\varphi}{\mathrm{d}z^2}\right), \qquad (5)$$

where we have taken $\mathbf{v}=v\hat{\mathbf{z}}$ and disregarded fluctuations. This equation can not be solved analytically for the model free energy functional of the LIPT ($f_0[\varphi]$ in equation 2), but a dimensional analysis of equation 5 predicts the interface width and speed to scale as $\delta \propto \sqrt{\kappa/h}$ and $v \propto \Gamma\sqrt{\kappa h}$, respectively. In addition, the projection of equation 5 onto the Goldstone mode $\mathrm{d}\varphi/\mathrm{d}z$ (ref. 29) also predicts a scaling $\sigma \propto (A-1)\sqrt{h}$ for the interfacial tension.

**Simulation details**. In all cases, numerical results are obtained from simulations of equation 3 under periodic boundary conditions, following usual stochastic algorithms for the noise field[38], and with a rescaled time step $\Gamma\Delta t = 5 \cdot 10^{-3}$.

Results in Fig. 3a,d,e are obtained on a $200 \times 200$ grid in simulation units $\sqrt{\kappa}$, and for $D = 0.02\kappa\Gamma$. A quench from $h = -2$ to $h = 0.25$ is applied at time $t = 0$. In Fig. 3d, the fit of the predicted scaling $v = b\Gamma\sqrt{\kappa h}$ to the simulation results gives the prefactor $b = 1.36 \pm 0.01$. In Fig. 3e, the simulation grid points contributing to the area covered by the metastable state are those featuring a value of the order parameter within its free energy basin. The limits of this basin are set by the inflection points of the local free energy (see Fig. 3b), which define the corresponding region of local thermodynamic stability, giving $\varphi \approx \pm 20°$.

In turn, simulations in Fig. 4b are performed on a $100 \times 100$ grid, for the same value of $D$, and the quench is applied from $h = 1$ to $h = -2$.

**Data availability**. The data that support the findings of this study are available from the corresponding authors upon request.

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

## Acknowledgements

R.A. acknowledges support from Fundació 'La Caixa', and from the University of Cambridge under the grant ITN-COMPLOIDS 234810 of the 7th Framework Programme of the EU. P.T. acknowledges support from the European Research Council under project No. 335040 and MINECO under project RYC-2011-07605. We acknowledge support from MINECO under project FIS2013-41144-P and Generalitat de Catalunya under project 2014-SGR-878.

## Author contributions

All authors conceived and planned the research. R.A. developed the theory and performed the simulations. P.T. performed the experiments. R.A. and P.T. analysed the experimental data. J.C. supervised the work. All authors discussed and interpreted results, and wrote the paper.

## Additional information

**Competing financial interests:** The authors declare no competing financial interests.

