## [Peer Review File · Nature Communications]

Reviewers' comments:

Reviewer #1 (Remarks to the Author):

This paper addresses an interesting question of quenching a colloidal crystal into an unstable state and observing its spinodal decomposition where another metastable phase is found. Solid-solid phase transformations are rich and timely, hence the topic concerns actual hot research in soft matter science and statistical physics. The implications are general. The paper is a nice combination of experiment on the particle-resolved level and theory and it systematically exploits the idea to use an external magnetic field to manipulate the system interactions.

I am inclined to recommend publication in Nature Communication but the authors should carefully amend the paper by clarifying the following points:

- 1) Figure 2: The scatter in the data concerning Figure 2b and 2c is pretty high. Nevertheless there seems to be an underlying deterministic oscillatory behavior in time. Is this physical or pure statistical error?
- 2) Page 2: Why is there an exact landscape inversion induced by the magnetic field? This needs to be much better explained rather than referencing to previous work.
- 3) There is a lot of work where spinodal decomposition happens into a disordered metastable glassy phase e.g. by Sciortino and coworkers, see for example G. Foffi et al, Arrested phase separation in a short-ranged attractive colloidal system: A numerical study, JCP 122, 224903 (2005). There are also corresponding experiments on colloid-polymer mixtures, see e.g. Zaccarelli et al, Gelation as arrested phase separation in short-ranged attractive colloid-polymer mixtures, JPCM 20, 494242 (2008). Hence the statement on page 1 that "metastable phases have never been observed to appear spontaneously by spinodal decomposition" seems to be too far fetched. See also the related work of Sappelt et al, Computer simulation study of phase separation in a binary mixture with a glass-forming component, Physica A 240, 453 (1997). A discussion of these works should be done by the authors and the message should be toned down.
- 4) The first two sentences of the abstract are not clear both for a general reader and the specialist. Please amend what the general message is.

Reviewer #2 (Remarks to the Author):

This manuscript reported novel scenarios of phase transitions driven by a non-conventional way (landscape inversion by tuning magnetic field). The metastable phases formed by spinodal decomposition are well explained in theory and quantified in simulations. Such non-standard way to drive phase transitions could be a good future research area with rich phenomena. The paper can be published if the following points can be improved.

The experimental results in fig.1 do not have the resolution to demonstrate the main point as described in the title of the paper. The authors should add more experimental images like Fig.1a,b or provide videos to illustrate the key point of the paper.

Page 1 "...form of a two-stage, liquid-mediated nucleation 11-13."

Ref. 11 is not about this. Nature Communications 7, 11113 (2016) confirmed the two-stage, liquid-mediated nucleation in an atomic system, which could be cited.

Page 3: "...either through nucleation or via specific mechanisms such as martensitic rearrangements in solids." should be something like "...either through diffusive nucleation or martensitic transformation" because martensitic transformations also follow nucleation processes although nuclei often grow rapidly e.g. at the speed of sound.

Some parts of the paper should be more clearly written and some sentences are difficult to read. For example, that $\alpha=0$ is a metastable phase should be introduced earlier. Fig.2a legend should be " $\alpha=\alpha_a$ & $\alpha=0$ coexistence", not α_a divided by α coexistence"

Page 6: "(data not shown)." Why not show the data?

Eq.4 misses a factor of $1/2$?

Supplementary eq.2 is for second order phase transition, which seems not appropriate to describe liquid-solid transitions.

The model is only for mean-field case. What if the noise is large?

Reviewer #3 (Remarks to the Author):

The manuscript by Alert and colleagues reports a combined experimental, theoretical and numerical analysis of the decomposition process following a instantaneous change of the external control parameter in a system of dipolar colloids. Depending on the value of the external magnetic field, the stable crystal structures are different (and with different degeneracy) allowing for the possibility of generating out-of-equilibrium scenarios which are not common in standard atomic and molecular systems. Specifically, it is possible to "jump" simultaneously to a metastable crystal coexisting (for a limited amount of time) with a stable crystal or start from a coexisting metastable and stable crystal and evolves with two different interfaces toward the new stable structure.

The article is clearly written, interesting, rich. Still, I find that the manuscript is too much oriented toward a theoretical modeling of the phenomenon, even if the group has a strong experimental expertise in realizing and investigating the real system. The experimental data are limited to Fig.2 and to a fraction of Fig.1. In addition, there is no attempt to closely compare theory and experiments. I would have expected a more accurate analysis of the experimental data, for example based on (for example) Steinhardt-like order parameter. An analysis based on spherical harmonics (the Q_n and/or W_n) could allow for a classification of the particles into different locally ordered structures. The time dependence of these quantities could help supporting the evidence in favor of the metastable crystal phase.

The interesting inverse "jump" where two distinct interfaces are observed is the result of a numerical study. I wonder if the authors are able of observing this phenomenon also in their experimental system. The only comment right now is that predictions "remain to be experimentally verified". But why is it ? To split the main findings in different publications ? Are there some additional difficulties in observing these new phenomena ?

In summary, a revised version of the manuscript enhancing the analysis of the experimental data (order parameters, scaling law and so on) and a closer comparisons between experiments and theory should be accepted in Nature Communications.

REVIEWERS' COMMENTS:

Reviewer #1 (Remarks to the Author):

The authors have revised the paper considering all points raised. I recommend publication in Nature Comm of the paper in its present form.

Reviewer #2 (Remarks to the Author):

My questions have been well replied. I recommend the publication.

Reviewer #3 (Remarks to the Author):

The revised version of the manuscript significantly improves over the original submission. It properly answers to all raised criticisms/suggestions.
I suggest publication of this work in its present form.

Reply to the referees' comments

Referee #1

This paper addresses an interesting question of quenching a colloidal crystal into an unstable state and observing its spinodal decomposition where another metastable phase is found. Solid-solid phase transformations are rich and timely, hence the topic concerns actual hot research in soft matter science and statistical physics. The implications are general. The paper is a nice combination of experiment on the particle-resolved level and theory and it systematically exploits the idea to use an external magnetic field to manipulate the system interactions.

I am inclined to recommend publication in Nature Communication but the authors should carefully amend the paper by clarifying the following points:

1. Figure 2: The scatter in the data concerning Figure 2b and 2c is pretty high. Nevertheless there seems to be an underlying deterministic oscillatory behavior in time. Is this physical or pure statistical error?

We have performed more experiments to reduce the scatter and strengthen our results (see new Fig. 2). This has clarified that the oscillatory trend was statistical error.

2. Page 2: Why is there an exact landscape inversion induced by the magnetic field? This needs to be much better explained rather than referencing to previous work.

This is now explained in the text. The caption of Fig. 1 is also expanded to give more details of the experimental system.

3. There is a lot of work where spinodal decomposition happens into a disordered metastable glassy phase e.g. by Sciortino and coworkers, see for example G. Foffi et al, Arrested phase separation in a short-ranged attractive colloidal system: A numerical study, JCP 122, 224903 (2005). There are also corresponding experiments on colloid-polymer mixtures, see e.g. Zaccarelli et al, Gelation as arrested phase separation in short-ranged attractive colloid-polymer mixtures, JPCM 20, 494242 (2008). Hence the statement on page 1 that "metastable phases have never been observed to appear spontaneously by spinodal decomposition" seems to be too far fetched. See also the related work of Sappelt et al, Computer simulation study of phase separation in a binary mixture with a glass-forming component, Physica A 240, 453 (1997).

A discussion of these works should be done by the authors and the message should be toned down.

We thank the referee for this comment. We now comment on this work and explicitly mention that we refer to the formation of metastable *equilibrium* phases, in the thermodynamic sense of the term, implying that such phases may be in a metastable state but correspond to actual equilibrium phases of the system for appropriate parameter values. This definition thus excludes other kinds of metastable states such as arrested states. We have implemented several small changes throughout the text, mainly in the introduction, to clarify this point.

The arrest of spinodal decomposition that the referee mentions gives rise to a nonequilibrium metastable state (a colloidal gel). We note that this is different from our result, which shows the formation of a metastable equilibrium phase by spinodal decomposition. To stress the difference, we note that, since it corresponds to an equilibrium phase, our metastable state (the rectangular lattice structure) can become the actual globally stable state by changing the appropriate control parameters. This was shown in Fig. 1b of Ref. [1]; the rectangular structure becomes the stable one at high densities (low $\langle d \rangle$). This is not possible for an arrested state such as the aforementioned gels. These are never the globally stable state because they do not correspond to equilibrium phases in the thermodynamic sense.

Another difference occurs during the phase separation process: our metastable phase starts forming from the very beginning of the spinodal decomposition; it forms directly from it. In contrast, the aforementioned gels only form upon the dynamical arrest that interrupts the spinodal decomposition. Finally, the formation of our metastable phase is generic, independent of the quench rate and depth. This also contrasts with the formation of the gel which, being due to a dynamic process, depends on the quench protocol.

As the referee points out, there is a lot of work on arrested spinodal decomposition, within different contexts and viewpoints. We believe that the works that are most related to ours are the ones on colloidal gelation [2–12]. All these works were reviewed in [9], which discusses the possible routes to colloidal gelation. The definitive confirmation of the arrested spinodal decomposition scenario was published shortly after [10]. Therefore, we have chosen to cite these two works.

4. The first two sentences of the abstract are not clear both for a general reader and the specialist. Please amend what the general message is.

We have rewritten the first half of the abstract to make our main point clearer.

Referee #2

This manuscript reported novel scenarios of phase transitions driven by a non-conventional way (landscape inversion by tuning magnetic field). The metastable phases formed by spinodal decomposition are well explained in theory and quantified in simulations. Such non-standard way to drive phase transitions could be a good future research area with rich phenomena. The paper can be published if the following points can be improved.

1. The experimental results in fig.1 do not have the resolution to demonstrate the main point as described in the title of the paper. The authors should add more experimental images like Fig.1a,b or provide videos to illustrate the key point of the paper.

We did not provide snapshots or videos of the colloidal crystal during the transition because it is very hard to identify well-defined domains of the different crystalline orders. Several factors contribute to this difficulty:

- For the particle density used in the experiments, $\alpha_b \approx 7^\circ$. Therefore, the α_b structure is very difficult to visually distinguish from the metastable structure with $\alpha_m = 0$.

- In the experimental conditions, the metastable potential well is very shallow, and hence the metastable domains are very small and short-lived.
- The lack of precise experimental control of the in-line particle density introduces variability in the equilibrium lattice angles and generates vacancies in the crystals, making the identification of domains more difficult.

However, interpreted by the model, the evidence based on the radial distribution function $g(r)$ is robust and conclusive. This is because all the small domains of the metastable phase, which are difficult to identify visually, contribute to the secondary peak of the $g(r)$. Hence, the effect is clearly reported by this structural observable.

We have performed more experiments that have substantially improved our results (see new Fig. 2). Averaging over 15 realizations, we have notably reduced the scatter in the data, and the effect can now be seen in another peak as well. The fourth resolved peak of the $g(r)$ also transiently increases during the phase separation, strengthening the evidence of the formation of the metastable phase.

2. Page 1 "...form of a two-stage, liquid-mediated nucleation 11-13." Ref. 11 is not about this. Nature Communications 7, 11113 (2016) confirmed the two-stage, liquid-mediated nucleation in an atomic system, which could be cited.

We agree with the referee that the citation was imprecise. We were citing Ref. [13] here to acknowledge their observation of a diffusive transformation in a colloidal crystal, even if it was not through melting. We now avoid this imprecision.

Ref. [14] came out after our submission. We knew about it and planned to cite it, which we now do.

3. Page 3: "...either through nucleation or via specific mechanisms such as martensitic rearrangements in solids." should be something like "...either through diffusive nucleation or martensitic transformation" because martensitic transformations also follow nucleation processes although nuclei often grow rapidly e.g. at the speed of sound.

We thank the referee for this comment. We have amended this.

4. Some parts of the paper should be more clearly written and some sentences are difficult to read. For example, that $\alpha=0$ is a metastable phase should be introduced earlier. Fig.2a legend should be " $\alpha = \alpha_a$ & $\alpha = 0$ coexistence", not α_a divided by α coexistence"

We have rephrased the description and interpretation of the experimental results to try to make it clearer. We now also designate the metastable phase by α_m to avoid confusion, and we introduce it earlier, already in Fig. 1c, in the explanation of the experimental results, and in the legend of Fig. 2a.

5. Page 6: "(data not shown)." Why not show the data?

We now provide a Supplementary Video showing this.

6. Eq.4 misses a factor of 1/2?

We thank the referee for his/her attention to the detail. A factor 1/2 was missing in the q^2 term of the dispersion relation, Eq. 4. We have now corrected it along with the corresponding Fig. 3c.

7. Supplementary eq.2 is for second order phase transition, which seems not appropriate to describe liquid-solid transitions.

We agree with the referee that Supplementary Eq. 2 does not correctly describe a possible liquid-solid transition in the system. However, here we used the simplest free energy that allowed us to stabilize liquid or crystalline phases by means of a parameter (a) and, at the same time, independently couple the crystalline phases to the LIPT scenario. The free energy in Eq. 2 serves this sole purpose, and allows us to show that the possible presence of liquid regions does not preclude the existence of two different interfaces. We now specify this in the Supplementary Information. Writing a free energy that correctly captures the details of the transition between the modulated liquid and the crystals in our 2D system might be possible but falls out of the scope of the present manuscript.

8. The model is only for mean-field case. What if the noise is large?

The dynamics incorporates noise (see Eq. 3). In the limit of large noise, the crystals would melt. In experiments, the ratio of thermal and magnetic energy scales is small, $k_B T/u_0 \approx 1.5 \times 10^{-4}$, and hence the effect of thermal fluctuations is weak. Instead, the main sources of noise in experiments are vacancies and inhomogeneities in the in-line particle density due to the presence of defects on the substrate, etc.

The effect of thermal fluctuations in the equilibrium order parameter of the crystals was characterized in Fig. S3 of [1]. There, we tuned the relative intensity of noise by modifying the magnetic field of the substrate, which modifies the strength of the dipolar interaction between the particles. This showed how stronger magnetic fields yield values of the order parameter closer to the mean-field predictions, as corresponds to a reduced effect of fluctuations.

Referee #3

The manuscript by Alert and colleagues reports a combined experimental, theoretical and numerical analysis of the decomposition process following a instantaneous change of the external control parameter in a system of dipolar colloids. Depending on the value of the external magnetic field, the stable crystal structures are different (and with different degeneracy) allowing for the possibility of generating out-of-equilibrium scenarios which are not common in standard atomic and molecular systems. Specifically, it is possible to "jump" simultaneously to a metastable crystal coexisting (for a limited amount of time) with a stable crystal or start from a coexisting metastable and stable crystal and evolves with two different interfaces toward the new stable structure.

1. The article is clearly written, interesting, rich. Still, I find that the manuscript is too much oriented toward a theoretical modeling of the phenomenon, even if the group has a strong experimental expertise in realizing and investigating the real system. The experimental data are limited to

Fig.2 and to a fraction of Fig.1. In addition, there is no attempt to closely compare theory and experiments. I would have expected a more accurate analysis of the experimental data, for example based on (for example) Steinhardt-like order parameter. An analysis based on spherical harmonics (the Q_n and/or W_n) could allow for a classification of the particles into different locally ordered structures. The time dependence of these quantities could help supporting the evidence in favor of the metastable crystal phase.

We have performed a new series of additional experiments to improve the statistics and strengthen our experimental results. Averaging over 15 realizations, we have notably reduced the scatter in the data in Fig. 2. In particular, this now allows to observe the effect in two peaks of the $g(r)$. Both the second and fourth peaks transiently increase during the phase separation, thus further strengthening the evidence for the formation of the metastable phase.

In addition, following the suggestion of the referee, we have analyzed the problem by means of different order parameters: the lattice angle α and the bond orientational order parameter for the square lattice, ψ_4 . The first one was introduced and discussed in Ref. [1]. Since particles are constrained in lines in our system, particles on the same line always for a zero bond angle regardless of the crystalline order, thus contributing an uninformative constant value to ψ_4 . Therefore, ψ_4 needs only to sum over neighbour particles in consecutive lines,

$$\psi_4 = \frac{1}{2} \sum_{j=1}^2 e^{i4\theta_j}, \quad (1)$$

where θ_j is the angle defined by the bond with particle j with respect to the direction of the lines. Hence, in terms of the lattice angle α , $\psi_4 = \cos(4\alpha)$.

While these quantities are generally appropriate to characterize the different phases, the particular angle values in our experiments are such that this characterization is not very helpful. Fig. 1 shows the evolution of the average values of α and $\text{Re}\{\psi_4\}$. Neither one can resolve the formation of the transient metastable phase, which would be indicated by a non-monotonic evolution. This is because the spatial average is dominated by the initial and final phases. Note that the initial phase has a very small angle $\alpha_b \approx 7^\circ$, so that ψ_4 is very close to 1, and hence cannot increase significantly. On the other hand, the (negative) contribution of the final phase is much larger, thus preventing to detect the possible increase of ψ_4 . A similar argument applies to the order parameter defined by the lattice angle. In contrast, the secondary peaks of the $g(r)$ correspond to only one of the phase-separating structures. Hence, the secondary peaks of the $g(r)$ allow to single out the formation of the metastable α_m structure. This is why we chose this structural observable instead of the evolution of the order parameter to monitor the phenomenon.

The order-parameter characterization may also be used locally to identify domains of different phases. We have tried to classify the particles into different local structures, as suggested by the referee. We have coloured each particle according to its local value of the lattice angle α or the bond orientational square order ψ_4 using the following criteria:

Figure 1: Evolution of the lattice angle α (left) and the real part of the bond orientational order parameter for the square lattice $Re\{\psi_4\}$ (right). Colour lines indicate the predicted equilibrium values for the initial and the final phases.

Figure 2: Snapshots of the system before, right after, and well after the quench. Particles are coloured according to their local values of the lattice angle α . The system initially forms domains of the two degenerated phases with angles $-\alpha_b$ and α_b (blue), separated by interfacial regions (white). Right after the quench, the system phase separates into small domains of the stable (black) and metastable (white) phases. Finally, metastable domains are eliminated and the system crystallizes in the homogeneous stable phase (black). Particles close to the borders of the image are not coloured to avoid boundary problems.

Figure 3: The same as Fig. 2 but with particles coloured according to their local values of the real part of the bond orientational square order parameter, $Re\{\psi_4\}$.

- Initial α_b structure in blue: those particles with local angle $(\alpha_m + \alpha_b)/2 < |\alpha| < (\alpha_a + \alpha_b)/2$, and the corresponding values of $\text{Re}\{\psi_4\}$.
- Metastable α_m structure in white: those particles with local angle $|\alpha| < (\alpha_m + \alpha_b)/2$, and the corresponding values of $\text{Re}\{\psi_4\}$.
- Final stable α_a structure in black: those particles with local angle $|\alpha| > (\alpha_a + \alpha_b)/2$, and the corresponding values of $\text{Re}\{\psi_4\}$.

Snapshots of the system before the quench, during the phase separation process, and in the final state are shown in Figs. 2 and 3, where colours are assigned according to the local values of α and $\text{Re}\{\psi_4\}$, respectively. Both procedures yield essentially equivalent results.

In the initial state, the system is in the α_b structure (blue). It is important to note that this structure is degenerated, so that crystals with $-\alpha_b$ and α_b angles are expected to coexist (see Fig. 1c in the paper). In fact, crystalline domains of opposed angles can be seen in the images. Then, the small white regions seem to be mainly interfaces (similar to grain boundaries) between domains of these two degenerate phases. Finally, isolated black particles are defects due to particle density inhomogeneities.

Right after the quench, the initial phase (blue) becomes unstable and starts to phase separate into small domains of the stable (black) and metastable (white) phases. Metastable domains are very small and short-lived. This is essentially due to two reasons:

- For the particle density in the experiments, the metastable potential well is very shallow.
- In the experimental conditions, $\alpha_b \approx 7^\circ$, and hence it is difficult to distinguish from the metastable structure with $\alpha_m = 0$.

Thus, clearly identifying large metastable domains is not possible in the current experimental setup since the particle density along the lines where the particles are allowed to move can not be well controlled, and hence the aforementioned issues can not be avoided. Indeed the lattice contains a significant number of vacancies which distort the analysis based on a local order parameter. Nevertheless, the secondary peaks of the $g(r)$ add the contributions of all the small metastable domains scattered through the system, and hence robustly report their formation. The new data strengthen the experimental validation of the effect, whose interpretation is clear in the light of the model.

Finally, the system reaches a homogeneous α_a structure phase (black).

2. The interesting inverse "jump" where two distinct interfaces are observed is the result of a numerical study. I wonder if the authors are able of observing this phenomenon also in their experimental system. The only comment right now is that predictions "remain to be experimentally verified". But why is it? To split the main findings in different publications? Are there some additional difficulties in observing these new phenomena?

Unfortunately, we can not experimentally observe this phenomenon due to the limitations and specificities of our experimental setup. We detail the main difficulties below, which we now also briefly mention in the text:

- Due to the relative abundance of vacancies, particles can easily move along the lines defined by the domain walls of the substrate. This allows for abrupt changes of angle at the contacts between the two degenerated phases, without a smooth crystalline interface between them. Knowing this, we built an extended model of our system that allows for the possibility of liquid-like interfaces (see Supplementary Information). This model shows that the presence of liquid interfaces does not preclude the existence of two different interfaces. However, in our experimental system, due to the particle discreteness and mobility at the scale of the domain sizes, the observation of smooth interfaces and their dynamics is not yet possible. This would require much larger system and domain sizes, and a significant reduction of the vacancies with a better control of the in-line particle density. These requirements are not accessible in our current setup.
- Even if smooth interfaces could be clearly observed in our system, the formation of the two different interfaces requires quenching the system during the coexistence of large domains of the stable and metastable phases. Then, spinodal decomposition occurring within stable-phase domains yields A-type interfaces, and that occurring within metastable-phase domains yields B-type interfaces (see Fig. 4 in the paper). The particle density in the experiments corresponds to a shallow metastable state in the energy, and hence the metastable domains are relatively small and short-lived. As a consequence, the coexistence of large enough stable and metastable domains is very difficult to achieve in our specific setup.

For these reasons, we can not verify the existence of the two different interfaces in our experimental setup. However, this phenomenon is a generic prediction of the LIPT scenario, that goes beyond our specific experimental realization of it. Therefore, it might be verified in the future in other experimental systems featuring LIPT, of which there is already one example [15]. Thus, instead of focusing only on the main result, we preferred to be more comprehensive and present the range of potential new phenomena associated to the phase-transition dynamics of the LIPT. Accordingly, we judged particularly interesting to explain the prediction of the two interfaces, illustrating it by means of numerical simulations. In addition, we also posed other appealing yet speculative questions that arise naturally from the LIPT, such as the open possibility of this scenario for conserved order parameter, with the intention to trigger further research in the community.

In summary, a revised version of the manuscript enhancing the analysis of the experimental data (order parameters, scaling law and so on) and a closer comparisons between experiments and theory should be accepted in Nature Communications.

We now include a stronger evidence for the formation of the metastable phase based on more statistics for the experimental data on the $g(r)$. We have performed the analyses suggested by the referee and explained why, unfortunately, the proposed observables are not good reporters of the metastable phase formation in our system whereas the $g(r)$ is. We have also explained the main difficulties that preclude the observation of the two different interfaces in our experimental setup. Thus, the revised manuscript features the closest comparison between predictions and experiments that can be reasonably achieved with our experimental setup.

List of changes

- Abstract: The first half of the abstract has been rewritten to make it more focused and clear.
- Page 1: A number of small changes throughout the introduction have been implemented to explicitly specify that we refer to the formation of metastable states of possible equilibrium phases of the system, in the thermodynamic sense. In particular, a sentence has been added to comment on the formation of nonequilibrium metastable states such as gels by dynamic arrests of spinodal decomposition processes.
- Page 1, column 2: The terminology on diffusive nucleation and displacive transformations in colloidal crystals has been amended. A reference to the observation of liquid-mediated nucleation in metals has been added.
- Page 2: The first paragraph of the “The landscape-inversion phase transition” section and the caption of Fig. 1 have been expanded to explain the origin of the inversion of the energy landscape, which have required giving more details about the experimental system.
- Pages 2-3: The first two paragraphs of the “Formation of metastable domains by spinodal decomposition” section have been partially rewritten to better explain the experimental data and to clarify the results. In particular, we now introduce earlier the values of the order parameter α that correspond to each crystalline state. The caption of Fig. 2, and panel Fig. 1c have also been modified accordingly.
- Figure 2: This figure now includes the data of the extra experiments that we have performed in order to improve the statistics and reduce the scatter in the data.
- Page 4: In the last sentence of the section “Phase coexistence with two different interfaces”, we now refer to and provide a Supplementary Movie to show the disappearance of the most energetic interface.
- Figure 3: Panel c now plots the correct dispersion relation.
- Page 6: In the second paragraph of the “Discussion” section, we now mention the main experimental limitations that preclude the observation of interfaces in our system.
- Page 6: As required by the journal guidelines, we now provide a description of the experimental methods in the “Methods” section, even if identical to those previously published.
- Page 7: A missing factor of $1/2$ has been corrected in Eq. 4.
- Other small changes throughout the text to accommodate the rest.
- Formatting changes to adapt it to the Nature Communications format requirements.
- In the Supplementary Information file, we now better explain the purpose and limitations of the extended model introduced to account for liquid-like interfaces. The second paragraph in section III has been modified to this end.

References

- [1] R. Alert, J. Casademunt, and P. Tierno. *Landscape-Inversion Phase Transition in Dipolar Colloids: Tuning the Structure and Dynamics of 2D Crystals*. Phys. Rev. Lett. **113**, 198301 (2014)
- [2] D. Sappelt and J. Jäckle. *Computer simulation study of phase separation in a binary mixture with a glass-forming component*. Phys. A Stat. Mech. its Appl. **240**, 453 (1997)
- [3] D. Sappelt and J. Jäckle. *Spinodal decomposition with formation of a glassy phase*. Europhys. Lett. **37**, 13 (1997)
- [4] F. Sciortino, S. Mossa, E. Zaccarelli, and P. Tartaglia. *Equilibrium Cluster Phases and Low-Density Arrested Disordered States: The Role of Short-Range Attraction and Long-Range Repulsion*. Phys. Rev. Lett. **93**, 055701 (2004)
- [5] G. Foffi, C. D. Michele, F. Sciortino, and P. Tartaglia. *Scaling of Dynamics with the Range of Interaction in Short-Range Attractive Colloids*. Phys. Rev. Lett. **94**, 078301 (2005)
- [6] G. Foffi, C. De Michele, F. Sciortino, and P. Tartaglia. *Arrested phase separation in a short-ranged attractive colloidal system: A numerical study*. J. Chem. Phys. **122**, 224903 (2005)
- [7] S. Manley, H. M. Wyss, K. Miyazaki, J. C. Conrad, V. Trappe, L. J. Kaufman, D. R. Reichman, and D. A. Weitz. *Glasslike Arrest in Spinodal Decomposition as a Route to Colloidal Gelation*. Phys. Rev. Lett. **95**, 238302 (2005)
- [8] F. Cardinaux, T. Gibaud, A. Stradner, and P. Schurtenberger. *Interplay between Spinodal Decomposition and Glass Formation in Proteins Exhibiting Short-Range Attractions*. Phys. Rev. Lett. **99**, 118301 (2007)
- [9] E. Zaccarelli. *Colloidal gels: equilibrium and non-equilibrium routes*. J. Phys. Condens. Matter **19**, 323101 (2007)
- [10] P. J. Lu, E. Zaccarelli, F. Ciulla, A. B. Schofield, F. Sciortino, and D. A. Weitz. *Gelation of particles with short-range attraction*. Nature **453**, 499 (2008)
- [11] E. Zaccarelli, P. J. Lu, F. Ciulla, D. A. Weitz, and F. Sciortino. *Gelation as arrested phase separation in short-ranged attractive colloid-polymer mixtures*. J. Phys. Condens. Matter **20**, 494242 (2008)
- [12] P. J. Lu and D. A. Weitz. *Colloidal Particles: Crystals, Glasses, and Gels*. Annu. Rev. Condens. Matter Phys. **4**, 217 (2013)
- [13] P. S. Mohanty, P. Bagheri, S. Nöjd, A. Yethiraj, and P. Schurtenberger. *Multiple Path-Dependent Routes for Phase-Transition Kinetics in Thermoresponsive and Field-Responsive Ultrasoft Colloids*. Phys. Rev. X **5**, 011030 (2015)
- [14] S. Pogatscher, D. Leutenegger, J. E. K. Schawe, P. J. Uggowitzer, and J. F. Löffler. *Solid-solid phase transitions via melting in metals*. Nat. Commun. **7**, 11113 (2016)
- [15] H. Carstensen, V. Kapaklis, and M. Wolff. *Phase formation in colloidal systems with tunable interaction*. Phys. Rev. E **92**, 012303 (2015)